# Effect of Enamel Matrix Derivatives on Osteoclast Formation from PBMC of Periodontitis Patients and Healthy Individuals after Interaction with Activated Endothelial Cells

**DOI:** 10.3390/medicina57030269

**Published:** 2021-03-15

**Authors:** Gerlinde Durstberger, Phuong Quynh Nguyen, Verena Hohensinner, Peter Pietschmann, Xiaohui Rausch-Fan, Oleh Andrukhov

**Affiliations:** 1Division of Conservative Dentistry and Periodontology, University Clinic of Dentistry, Medical University of Vienna, 1090 Vienna, Austria; gerlinde.durstberger@meduniwien.ac.at (G.D.); xiaohui.rausch-fan@meduniwien.ac.at (X.R.-F.); 2Competence Center for Periodontal Research, University Clinic of Dentistry, Medical University of Vienna, 1090 Vienna, Austria; phuong.nguyen@meduniwien.ac.at; 3Institute of Pathophysiology and Allergy Research, Center of Pathophysiology, Infectiology and Immuology, Medical University of Vienna, 1090 Vienna, Austria; verena.hohensinner@bmi.gv.at (V.H.); peter.pietschmann@meduniwien.ac.at (P.P.)

**Keywords:** periodontitis, enamel matrix derivative, osteoclast

## Abstract

*Background and objectives:* Enamel matrix derivative (EMD) is produced from developing porcine tooth buds and represents a complex of low-molecular-weight hydrophobic enamel proteins. EMD is widely applied in periodontal regeneration. Osteoclasts are multinuclear cells, which are responsible for bone resorption. The precursors of osteoclasts, hematopoietic cells, undergo in vivo the process of transendothelial migration before differentiation. EMD is known to affect the process of osteoclastogenesis, but its effect on human osteoclasts precursors after the interaction with activated endothelium was never studied. *Materials and Methods:* Human umbilical vein endothelial cells (HUVECs)s were seeded in transwell inserts with a pore size of 8 µm and pre-activated by TNF-α and IL-1β for 18 h. Peripheral blood mononuclear cells (PBMCs), freshly isolated from 16 periodontitis patients and 16 healthy individuals, were added to pre-activated HUVECs. Adherent, non-adherent and transmigrated cells were collected and differentiated to osteoclasts by the standard protocol in the presence or absence of EMD. The number of osteoclasts was determined by tartrate-resistant acid phosphatase staining. *Results:* PBMCs isolated from periodontitis patients have formed a significantly higher osteoclast number compared to PBMCs isolated from healthy individuals (*p* < 0.05). EMD induced concentration-dependent inhibition of osteoclast formation from PBMCs. This was true for the different PBMC fractions isolated from both healthy individuals and periodontitis patients. *Conclusions:* Our data show that EMD inhibits the formation and activity of osteoclasts differentiated from the progenitor cells after the interaction with activated endothelium. This might be associated with bone resorption inhibition and supporting bone regeneration in the frame of periodontal therapy.

## 1. Introduction

Osteoclasts are multinuclear cells that originate from hematopoietic cells of the lineage of monocytes and macrophages, which are responsible for bone resorption [1]. Human osteoclast precursors are present in the peripheral blood mononuclear cells (PBMC) [2]. Differentiation of osteoclasts precursor cells into mature osteoclasts is driven by macrophage colony-stimulating factor (M-CSF) and receptor activator of nuclear factor-kB (RANKL) [3]. Differentiation of PBMC into osteoclasts is a crucial process associated with bone loss in periodontal disease [4,5]. Clinical studies show an increased number of osteoclasts in the lesion sites in patients with advanced periodontitis. Several previous studies suggest that PBMC isolated from patients with chronic periodontitis might spontaneously differentiate into active osteoclasts even without additional stimulation with M-CSF and RANKL [6,7,8]. Some studies report an increased osteoclast formation from PBMC isolated from periodontitis patients compared to healthy individuals [6,8]. In contrast, one study reports a lower number of osteoclasts formed from PBMC of periodontitis patients upon stimulation with M-CSF and RANKL compared to healthy controls [7].

Osteoclast precursor cells in blood need to migrate through the endothelial barrier to the place of bone resorption. The migration of osteoclast precursors from circulating blood into tissue is regulated by the endothelium underlying the inner surface of blood vessels. Activation of endothelial cells by pro-inflammatory mediators, such as interleukin (IL)-1β and tumor necrosis factor (TNF)-α results in an increased expression of adhesion molecules by endothelial cells and promotes adhesion and transendothelial migration (TEM) of osteoclast precursors [9]. PBMCs after TEM exhibit enhanced osteoclast formation and bone-resorption activity upon stimulation with M-CSF and RANKL [10]. Both studies show that PBMCs, which do not adhere to activated endothelium, cannot form osteoclasts. The recruitment of osteoclast precursors by endothelial cells and subsequent osteoclasts formation might reflect processes taking place during bone destruction due to inflammatory diseases, such as periodontitis or periimplantitis.

Enamel matrix derivative (EMD) includes several low-molecular-weight hydrophobic enamel proteins (≤20 kDa) originating from developing porcine tooth buds. The EMD-based commercial product Emdogain (Straumann, Basel, Switzerland) has been successfully used in clinics for many years to promote periodontal regeneration [11,12]. The influence of EMD on biological processes seems to be based on the presence of bioactive compounds, particularly amelogenin and amelogenin peptides with a molecular weight of 5, 9 and 12 kDa [13]. These peptides are involved in the process of teeth development and, together with EMD’s ability to activate the TGF-β pathway, they account for the biological effects of EMD [14,15]. The effect of EMD on the formation of osteoclasts was investigated only in different small animal models and the results of these studies are rather controversial. Some studies show that EMD stimulates osteoclasts formation in mice in vitro [16,17]. The stimulation of osteoclast formation by EMD in the murine model seems to be mediated by the TGF-β pathway [18]. In contrast to these observations, another study shows that amelogenin, which is the main component of EMD, inhibited osteoclasts formation from murine bone marrow cells [19]. No effect of EMD on osteoclasts activity was found by application in rat femur [20]. However, to the best of our knowledge, to date, there is no study on the effect of EMD on osteoclast formation in humans. In the present study, we investigated the effect of EMD on osteoclast formation from PBMC of periodontitis patients and healthy individuals. Peripheral blood was isolated from periodontitis patients and healthy individuals. Human umbilical vein endothelial cells were activated by IL-1β and TNF-α and the osteoclast precursor cells were isolated through two different criteria: adherence to activated HUVECs and transendothelial migration. Osteoclast formation from osteoclast precursors was induced by incubation with MCF-S, RANKL and dexamethasone in the presence or absence of different concentrations of EMD. Osteoclast formation was quantified by tartrate-resistant acid phosphatase (TRAP) staining.

## 2. Materials and Methods

### 2.1. Patient Selection

The study protocol was approved by the Ethics Committee of the Medical University of Vienna (Protocol No.: 273/2006, approved on 23 July 2006). This cross-sectional study included 16 periodontitis patients (11 men and 5 women) and 16 periodontally healthy volunteers (10 men and 6 women). Clinical history was recorded for all participants (personal data and medical history). All participants were thoroughly informed about the aims and methods of the study and gave informed written consent. The patient group consisted of periodontitis patients recruited at the Division for Conservative Dentistry and Periodontology, University Clinic of Dentistry, Medical University of Vienna. The following exclusion criteria were applied: the presence of any systemic disease (e.g., diabetes mellitus, asthma and malignancies), acute infection, periodontal treatment within the last three months, immune-suppressive medication or immunodeficiency, heavy smokers (more than 10 cigarettes/day), xerostomia or any other disease of the salivary glands, less than 20 teeth, pregnancy or lactation, history of radio- or chemotherapy, intake of antibiotics, immunomodulatory and anti-inflammatory drugs during 3 months prior to the study. Every participant underwent a panoramic radiographic examination. Bone loss in the periodontitis groups was additionally evaluated with intra-oral radiographs. For periodontal diagnostics, among others, probing pocket depth (PPD), clinical Attachment Level (CAL) and bleeding on probing (BoP) were recorded at 6 sites per tooth by experienced periodontists at the Department of Periodontology. The periodontitis patients group included subjects with generalized (≥30% affected sites) periodontitis Stage III or IV (loss of supporting bone extending to the middle or apical third of the root) Grade A, B or C [21]. The control group was recruited from colleagues, relatives and acquaintances as well as periodontal healthy clinical patients. Periodontal health was confirmed according to the 2018 classification of periodontal health [22]. In the healthy group, periodontal screening index was 0, 1 or 2 and the intact alveolar bone hight was verified in an panoramic x-ray.

### 2.2. HUVECs Culture and PBMCs Isolation

Human umbilical vein endothelial cells (HUVECs) were cultured in endothelial cell medium (ECM) supplemented with 100 µg/mL streptomycin, 100 U/mL penicillin, 2 mM L-glutamine, 0.25 µg/mL fungizone, 5 U/mL heparin, 30–50 µg/mL endothelial cell growth supplement and 20% fetal calf serum (FCS) [13]. HUVECs were cultured in culture flasks coated with 0.2% gelatin at 37 °C in a humidified atmosphere of 5% CO_2_ and 95% air.

Whole blood (40 mL) was collected from patients with severe, generalized periodontal disease and healthy volunteers as controls. The vacutainers contained lithium heparin. Patients with systemic diseases and smokers were excluded. PBMCs were isolated by density gradient centrifugation and washed twice with HBSS and resuspended in α-MEM supplemented with 10% fetal calf serum (FCS), 2 mM L-glutamine, penicillin 100 U/mL and streptomycin 100 μg/mL (PBMC medium).

### 2.3. Isolation of Adherent and Non-Adherent Fractions

The experimental model of PBMC differentiation into osteoclasts after co-culture with activated endothelial cells was established in our laboratory. In these experiments, HUVECs were seeded at a density of 5 × 10^5^ cells per well in 10 mL of ECM in Petri dishes coated with 0.2% gelatin. 24 h after the seeding, HUVECs were pre-activated by 25 ng/mL human recombinant tumor necrosis factor (TNF)-α (PeproTech, Rocky Hill, NJ, USA) and 10 U/mL human recombinant interleukin (IL)-1β (PeproTech, Rocky Hill, NJ, USA) for 18 h. Afterward, HUVECs were rinsed twice with PBS and then 50 × 10^6^ freshly isolated PBMCs were added to each Petri dish. After 90 min incubation, non-adherent PBMC fractions were collected from the Petri dish. Adherent PBMC fraction was separated from endothelial cells by magnetic-activated cell sorting (MACS) by negative selection using human CD31 magnetic beads (Miltenyi Biotec, Bergisch Gladbach, Germany) to separate endothelial cells. Both non-adherent and adherent fractions of PBMCs were used for the generation of osteoclasts.

### 2.4. Trans-Endothelial Migration of PBMCs

The experiments on trans-endothelial migration of PBMCs were designed according to the previously described report [10]. Transwell inserts (8 µm, Sarstedt, Nürnbrecht, Germany) were pre-coated with 0.2% gelatin and 1 × 10^5^ HUVECs were seeded on them in 3ml of ECM medium. After 24 h, the media were changed into ECM media supplemented with 25 ng/mL TNF-α and 10 U/mL IL-1β (PeproTech, Rocky Hill, NJ, USA) and the cells were stimulated for 18 h. After stimulation, the inserts were washed twice with PBS, placed into 6 well plates and 5 × 10^6^ freshly isolated PBMCs were added. The migration was allowed to proceed for 3 h at 37 °C. Trans-migrated PBMCs were collected, pulled, counted and used for the osteoclasts differentiation assay.

### 2.5. Generation of Osteoclast-Like Cells from Different PBMCs Fraction

PBMC were plated out at 5 × 10^5^ cells/well in 96 well plates. Cells were cultured at 37 °C for 21 days in Osteoclast medium (PBMC medium supplemented with 30 ng/mL of RANKL, 25 ng/mL of macrophage-colony stimulating factor and 10 nM dexamethasone).

Lyophilised EMD powder (Straumann, Basel, Switzerland) was dissolved in 0.1% acetic acid to obtain the solution with a stock concentration of 10 mg/mL. A concentration series of EMD was prepared by diluting the stock solution to concentrations of 100 µg/mL, 10 µg/mL, 1 µg/mL, 100 ng/mL and 10 ng/mL with osteoclasts medium. Half of the medium was changed three times per week. Additionally, three controls were performed (A: PBMC-Medium, B: Osteoclast medium, C: Vehicle control with 0.01% acetic acid in osteoclast medium).

On day 21, osteoclast cell formation was determined. Adherent cells were fixed with a fixative solution for 10 min. After washing two times with a distilled water, cells were stained for tartrate-resistant acid phosphatase (TRAP) using a commercially available histochemical kit. TRAP^+^ multinuclear cells containing three or more nuclei were counted as osteoclasts.

### 2.6. Statistical Analysis

The Kolmogorov–Smirnov test was used to confirm a normal distribution. The differences in the number of osteoclasts between healthy individuals and periodontitis patients were analyzed using a t-test. In the experiments with the different EMD concentrations, the differences between groups were analyzed by ANOVA for repeated measures followed by the LSD-post-hoc test. The software SPSS 22.0 (IBM, Armonk, NY, USA) was used for the statistical analysis. *p* values < 0.05 were considered to be statistically significant. Data are presented as means ± SD. In the experiments with adherent cells, PBMCs were isolated from 8 healthy individuals and 9 periodontitis patients. In the experiments with transendothelial migration, PBMCs were isolated from 8 healthy individuals and 7 periodontitis patients.

## 3. Results

### 3.1. Study Participants’ Demographical Characteristics and Clinical Parameters

Clinical parameters of periodontitis patients and demographic characteristics of all study participants are summarized in Table 1.

### 3.2. Osteoclasts-Like Cells in Different PBMCs Fraction Isolated from Healthy Individuals and Periodontitis Patients

The exemplary TRAP staining and the number of TRAP^+^ multinucleated cells differentiated from the different fractions of PBMCs isolated from periodontitis patients and healthy individuals are shown in Figure 1 and Figure 2, respectively. Adherent and trans-migrated cells isolated from periodontitis patients exhibited a significantly higher number of osteoclast-like cells compared to healthy individuals. No difference in the number of osteoclast-like cells between periodontitis and control groups was observed for non-adherent PBMC fraction.

### 3.3. Effect of EMD on the Number of Osteoclast-Like Cells in Different PBMCs’ Fractions

The effect of different concentrations of EMD on the number of osteoclast-like cells differentiated from adherent, non-adherent and migrated fractions of PBMCs is shown in Figure 3, Figure 4 and Figure 5, respectively. In all fractions, EMD induced a dose-dependent decrease in the number of TRAP^+^ multinucleated cells. In HUVECs-adherent PBMCs, statistically significant differences compared to controls (control and vehicle control) were observed for 100 µg/mL of EMD in both periodontitis patients and healthy individuals. Additionally, in periodontitis patients, 1 µg/mL of EMD induced a significantly lower number of osteoclasts compared to the control. EMD at the concentration of 100 µg/mL induced a significant decrease in the number of osteoclast-like cells compared to both control and vehicle control in non-adherent PBMCs. In transmigrated PBMC, a significant decrease in the number of osteoclast-like cells was observed after treatment with 10–100 µg/mL of EMD in both periodontitis patients and healthy individuals.

## 4. Discussion

Osteoclasts play a crucial role in the bone loss observed in periodontitis, periimplantitis and other inflammatory diseases [23,24] and derive from cells of the monocyte lineage present in the bone marrow and peripheral blood [25]. Previous studies have shown that PBMCs isolated from the peripheral blood of periodontitis patients might differentiate in osteoclasts even in the absence of MCS-F and RANKL. Still, in the presence of these factors, no differences were observed between healthy and periodontitis groups [6,7]. Furthermore, osteoclasts derived from PBMCs of periodontitis patients exhibit higher bone resorption activity [6,7,8]. In our study, the differentiation of PBMCs into osteoclasts was induced in the presence of MCS-F and RANKL. However, in contrast to previous studies, the formation of osteoclasts in PBMCs of periodontitis patients was higher for the adherent and trans-migrated fractions of PBMCs. Only for the non-adherent PBMCs fractions, no differences between periodontitis patients and healthy individuals were observed. It should be noted that previous studies did not observe the osteoclast formation from non-adherent PBMCs’ fraction [9,10], which is in contrast to our data. This discrepancy can be explained by different types of endothelial cells as well as various activation protocols.

The difference in osteoclast formation in different PBMCs’ fractions can be explained by the fact that the interaction of PMBCs with activated endothelium changes their composition. Activation of endothelial cells with pro-inflammatory cytokines upregulates the expression of adhesion molecules, such as intercellular adhesion molecule 1 (ICAM-1), vascular adhesion molecule 1 (VCAM-1) and E-selectin [26]. These molecules regulate the adhesion of different PBMCs’ subsets to endothelial cells. Particularly, activated endothelial cells promote the adhesion of CD14^+^ monocytes, CD16^+^ natural killer cells, CD4^+^ CD45RA- memory T cells, but not naïve CD4^+^CD45^+^ T cells [27,28]. PBMCs after trans-endothelial migration also exhibit a higher proportion of CD14^+^ monocytic cells [10]. Different proportions of CD14^+^ cells might explain our observation that the absolute number of osteoclast-like cells was higher for adherent and trans-migrated PBMCs than for non-adherent PBMCs.

We have found that EMD inhibits the formation of osteoclast-like cells in a concentration-dependent manner. This finding contradicts the existing literature: previous studies show the stimulatory effect of EMD on osteoclasts formation [16,17]. The essential difference between our and previous studies is different cell sources. In our study, osteoclasts were generated from human PMBCs, whereas other studies used mouse bone marrow macrophages. The generation of osteoclasts from humans and mice requires different protocols [29], which might account for the different results.

The major component of EMD is amelogenin [30]. In addition, EMD contains TGF-β like activity [31,32]. Amelogenin is shown to inhibit the osteoclasts formation from murine bone marrow cells [19] and suppress root resorption, RANKL production and osteoclast formation in vivo in a rat model [33]. TGF-β is known to stimulate the formation of osteoclasts in mice [34], but a recent study has shown that TGF-β suppresses RANKL-induced human osteoclasts development from PBMCs [35]. Therefore, the suppressive effect of EMD on the formation of osteoclast-like cells observed in our study could be due to both amelogenin and TGF- β activity.

Interestingly, some quantitative differences in the effect of EMD were observed between the different PBMCs fractions. Particularly trans-migrated PBMCs were more sensitive to EMD compared to adherent and non-adherent fractions. In trans-migrated PBMCs, the inhibitory effect of EMD was observed starting from 1 µg/mL of EMD, whereas, in adherent and non-adherent fractions, the inhibitory effect was observed only at 100 µg/mL of EMD. This difference could be explained by the differences in the composition of different PBMCs fractions. In addition, trans-endothelial migration might alter the properties of PBMCs and their susceptibility to the different biologically active compounds. However, these assumptions need to be further confirmed experimentally.

Periodontitis is an inflammatory disease characterized by alveolar bone resorption and subsequent pocket formation. Osteoclasts are essential for periodontal bone resorption and their development and activity are regulated by numerous inflammation-associated factors, particularly RANKL and M-CSF. The goals of periodontal therapy are to eliminate inflammation, stop bone loss and achieve periodontal regeneration. EMD demonstrated its regenerative capacity for periodontal tissues and bone in many clinical and experimental studies [36], but its effects on osteoclasts and the RANKL- and OPG- pathways have not yet been investigated [16,37]. Our data show that EMD might inhibit bone resorption by inhibiting osteoclast formation, but the clinical relevance of this EMD effect should be further confirmed.

The major limitation of this study is its in vitro character and using a relatively low participant number. Osteoclasts were identified only based on the presence of at least three nuclei and positive TRAP staining, but we did not determine the bone-resorbing activity of osteoclasts. Such activity was not determined because of a rather limited number of PBMCs obtained after interaction with activated endothelium. The fusion of several macrophages can lead to the formation of multinucleated giant cells [38] and the bone-resorbing activity is a major factor to distinguish the osteoclasts from them [39]. However, since we used a common protocol for osteoclasts generation in the presence of RANKL and M-CSF, the presence of multinucleated giant cells or fused macrophages in our samples is not very likely.

## 5. Conclusions

In our study, we found that PBMCs isolated from periodontitis patients have formed a significantly higher number of osteoclast-like cells compared to PBMCs isolated from healthy individuals. Furthermore, our data show that EMD inhibits the formation of osteoclast-like cells differentiated from PBMCs cells after the interaction with activated endothelium in a dose-dependent manner. This additional osteoclast-inhibiting effect of EMD might be beneficial in the inhibition of bone resorption in the frame of periodontal therapy. However, the clinical relevance of this EMD effect should be confirmed in further in vivo studies.

## Figures and Tables

**Figure 1 medicina-57-00269-f001:**
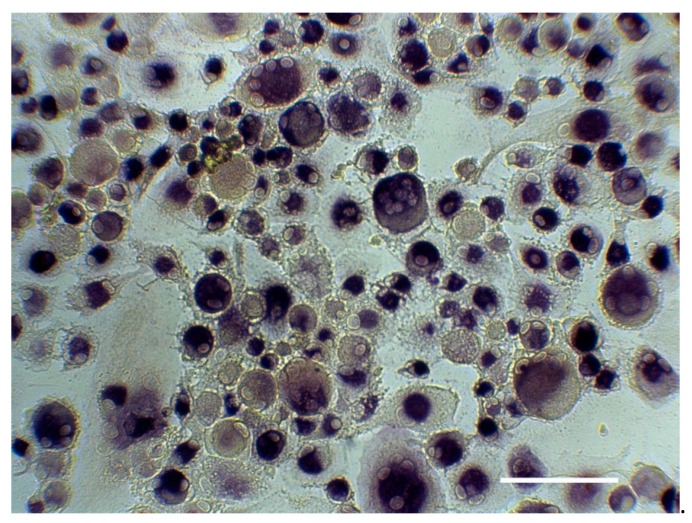
Exemplary tartrate-resistant acid phosphatase(TRAP) staining. Osteoclasts were differentiated from adherent PBMCs of periodontitis patient in the presence of RANKL, macrophage-colony stimulating factor and dexamethasone for 21 days and stained with TRAP. Positively stained cells with ≥3 nuclei were considered as osteoclasts. The photo has been taken under a microscope (Eclipse TS100, Nikon, Germany) using Optocam-II camera (Optoteam, Germany) at 20× magnification. Scale bar corresponds to 200 µm.

**Figure 2 medicina-57-00269-f002:**
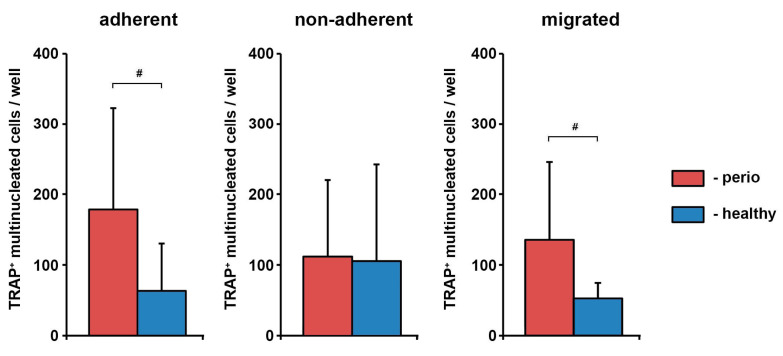
The number of TRAP^+^ multinucleated cells differentiated from the different fractions of peripheral blood mononuclear cells (PBMCs) isolated from periodontitis patients and healthy individuals. *Y*-axis represents the number of TRAP^+^ multinucleated cells. Data are presented as mean ± SD of 9 periodontitis patients and 8 healthy individuals (adherent, non-adherent fractions) or 7 periodontitis patients and 8 healthy individuals (migrated fractions). #-significantly different between periodontitis patients and healthy individuals.

**Figure 3 medicina-57-00269-f003:**
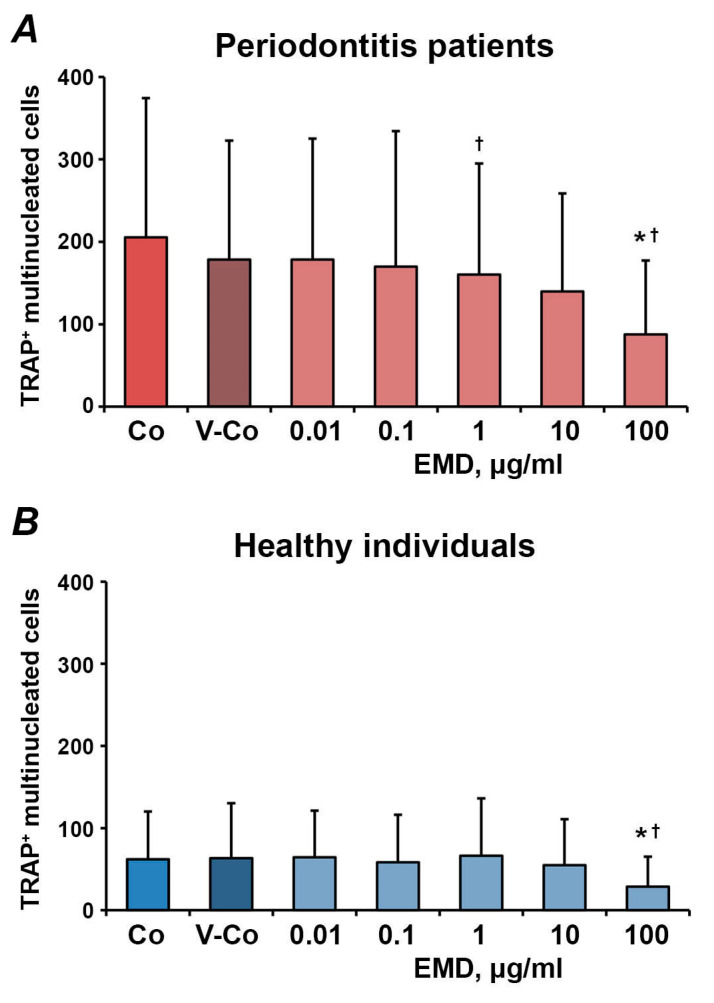
The effect of enamel matrix derivative (EMD) on the formation of osteoclast-like cells from human umbilical vein endothelial cells (HUVECs)-adherent PBMCs. HUVECs were pre-activated with 25 nM TNF-α and 10 U/mL IL-1β and co-cultured with freshly isolated PBMCs. A HUVECs-adherent fraction of PBMCs was isolated by magnetic separation and used for osteoclast-formation assay in the presence of different EMD concentrations. Cells treated with 0.01% of acetic acid were used as vehicle control (V-Co). Cells without EMD treatment were used as control (Co). *Y*-axis represents the number of TRAP^+^ multinucleated cells. Data are presented as mean ± SD of 9 periodontitis patients (**A**) and 8 healthy individuals (**B**). *-significantly different compared to V-Co; †-significantly different compared to Co.

**Figure 4 medicina-57-00269-f004:**
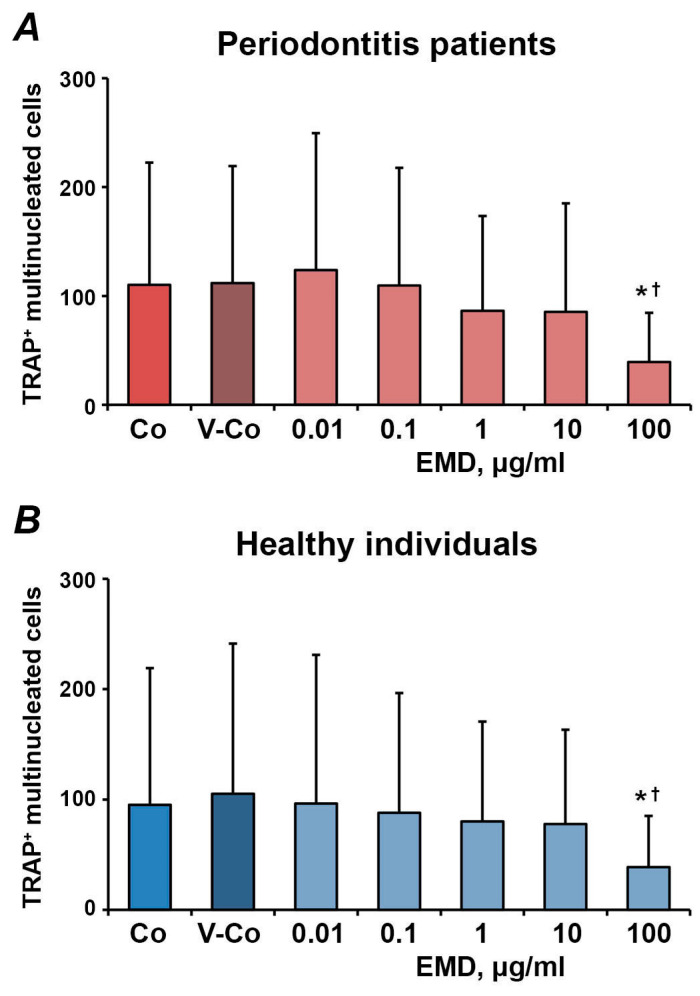
The effect of EMD on the formation of osteoclast-like cells from PBMCs non-adherent to pre-activated HUVECs. HUVECs were pre-activated with 25 nM TNF-α and 10 U/mL IL-1β and co-cultured with freshly isolated PBMCs. Non-adherent PBMCs were collected and used for osteoclast-formation assay in the presence of different EMD concentrations. Cells treated with 0.01% of acetic acid were used as vehicle control (V-Co). Cells without EMD treatment were used as control (Co). *Y*-axis represents the number of TRAP^+^ multinucleated cells. Data are presented as mean ± SD of 9 periodontitis patients (**A**) and 8 healthy individuals (**B**). *-significantly different compared to V-Co; †-significantly different compared to Co.

**Figure 5 medicina-57-00269-f005:**
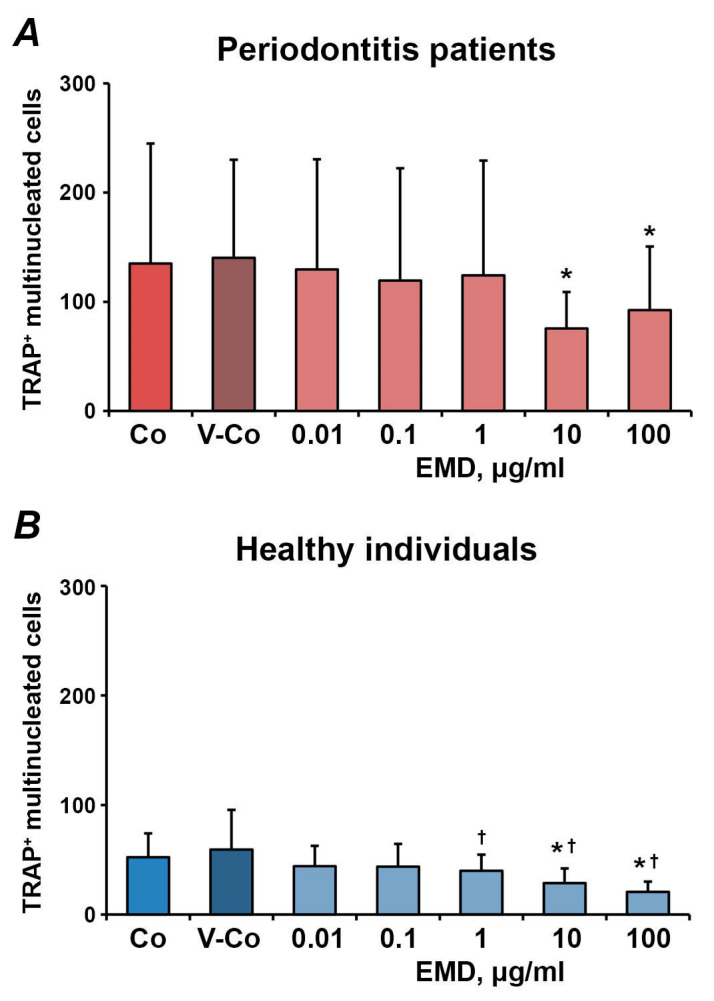
The effect of EMD on the formation of osteoclast-like cells from PBMCs after trans-endothelial migration. HUVECs were seeded on Transwell inserts with 8 µM pore size and pre-activated with 25 nM TNF-α and 10 U/mL IL-1β. Freshly isolated PBMCs were applied into Transwell inserts, transmigrated PBMCs were collected for 3 h and used for osteoclast-formation assay in the presence of different EMD concentrations. Cells treated with 0.01% of acetic acid were used as vehicle control (V-Co). Cells without EMD treatment were used as control (Co). *Y*-axis represents the number of TRAP^+^ multinucleated cells. Data are presented as mean ± SD of 7 periodontitis patients (**A**) and 8 healthy individuals (**B**). *-significantly different compared to V-Co; †-significantly different compared to Co.

**Table 1 medicina-57-00269-t001:** Demographic characteristics and clinical parameters of study groups. Data are presented as mean ± SD. TEM, trans-endothelial migration; PPD, probing pocket depth; BoP, bleeding on probing. PPD and BoP were calculated based on the data measured at 6 sites of each present tooth.

	Adhesion Experiments		TEM Experiments	
	Healthy, *n* = 8	Periodontitis, *n* = 9	Healthy, *n* = 8	Periodontitis, *n* = 7
Age, years	37.5 ± 10.5	43.8 ± 7.0	45.1 ± 9.7	48.7 ± 6.0
Gender, m/f	5/3	7/2	5/3	4/3
PPD, mean, mm		3.66 ± 0.49		3.60 ± 1.10
PPD, range, mm	0–3	0–10	0–3	0–12
BoP, %		35.8 ± 27.2		45.0 ± 32.9

## Data Availability

The data presented in this study are available on request from the corresponding author.

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
