# Peer review of "Effect of Enamel Matrix Derivatives on Osteoclast Formation from PBMC of Periodontitis Patients and Healthy Individuals after Interaction with Activated Endothelial Cells"

_medicina, 2021, doi:10.3390/medicina57030269_

Round 1

Reviewer 1 Report

The research work carried out by the authors is very interesting and is well developed. There are some grammatical errors throughout the manuscript. The discussion should be expanded to included a weaknesses paragraph. The discussion and conclusions are rather flat.

Author Response

COMMENT 1:

The research work carried out by the authors is very interesting and is well developed. There are some grammatical errors throughout the manuscript. The discussion should be expanded to included a weaknesses paragraph. The discussion and conclusions are rather flat.

RESPONSE;

Thank you very much for your positive and constructive feedback. Please, find below our response to the raised criticism points. The discussion was revised, and the paragraph about study limitations was extended (lines 315-324). We also carefully checked the manuscript to remove the grammatical errors and typos. All changes are highlighted in yellow.

Reviewer 2 Report

Thank you for the submission of this interesting high-quality paper. However, i made some comments to the author for the improvement of the article. 

Abstract: 

L. 26: Please explain the abbreviation TRAP with its first appearance.

Please state early in the abstract, how many healthy patients and periodontitis patients were included to sample collection. 

Introduction: 

L. 37: Please write "lineage of monocytes and macrophages" instead of "monocyte/macrophage lineage".

L. 42: "PMC": Please be consistent with the abbreviations or give an explanation for PMC. Only PBMC was used before. 

L. 63: Please provide a definition of "low molecular weight".

L. 65: Please remove the Trademark from Emdogain and provide manufacturer and location informations in brackets (Straumann...)

L. 68: Please give a definition or at least an example for "bioactive compounds" 

Materials&Methods: 

Different progression grades according to the new periodontitis classification stand for different host responses. Have you investigated the influence of the grades further?

L. 168: Please use the commonly used SD instead of S.E.M. 

Results:

Table 1: In the legend, you use the abbreviation PPD, in the table PD. Please correct. To the reviewer it is not clear, why you do not provide BOP and PD of healthy patients. 

Figure 1: Please give the reader the information, if this exemplary picture was obtained from a healthy or periodontitis patient. I would furthermore suggest to provide both, a healthy and a periodontitis example. Can you depict an endothelial cell as well in this exemplary picture?

You should, either in the M&M section or at least in the Figure legend, give some information about the micrscope and the used settings. 

Fig. 4: Wouldn't a comparison with Co be more relevant compared to V-Co?

Discussion: In the M&M sections, you write about patients with severe, generalized periodontal disease. Please discuss the quite low mean PPD of 3.6mm.

Please discuss, how you made sure, that you see osteoclasts and no other cells, e.g. macrophages, in your wells. Have you performed some western blot or ELISA? 

Line 294: As above: I would not seperate RANKL and OPG with a slash. I would suggest "RANKL- and OPG-pathway"

Author Response

Thank you very much for your positive and constructive feedback. Please, find below our response to the raised criticism points.

COMMENT 1:

  1. 26: Please explain the abbreviation TRAP with its first appearance.

Please state early in the abstract, how many healthy patients and periodontitis patients were included to sample collection. 

RESPONSE:

The abstract was modified according to the Reviewer's suggestion (see text highlighted in yellow).

COMMENT 2:

Introduction: 

  1. 37: Please write "lineage of monocytes and macrophages" instead of "monocyte/macrophage lineage".
  2. 42: "PMC": Please be consistent with the abbreviations or give an explanation for PMC. Only PBMC was used before.
  3. 63: Please provide a definition of "low molecular weight".
  4. 65: Please remove the Trademark from Emdogain and provide manufacturer and location informations in brackets (Straumann...)
  5. 68: Please give a definition or at least an example for "bioactive compounds"

RESPONSE:

The first and second points were corrected according to the Reviewer's suggestion. EMD includes proteins with a molecular weight less than 20 kDa; this is mentioned in the revised version. Trademark was removed, and information about Straumann was added (lines 65-66). We also provided some examples of bioactive compounds in EMD (lines 68-72).

COMMENT 3

Materials&Methods: 

Different progression grades according to the new periodontitis classification stand for different host responses. Have you investigated the influence of the grades further?

RESPONSE:

We did not investigate the effect of different grades because of a relatively low number of participants.

COMMENT 4:

  1. 168: Please use the commonly used SD instead of SEM.

RESPONSE:

We have changed our data presentation and use SD instead of SEM Figures 2-5 were revised.

COMMENT 5:

Results:

Table 1: In the legend, you use the abbreviation PPD, in the table PD. Please correct. To the Reviewer it is not clear, why you do not provide BOP and PD of healthy patients. 

RESPONSE:

Periodontal health was confirmed by assessing a periodontal screening index and x-rays, a complete periodontal examination only was performed in the patient group. Therefore, the parameters PPD and BoP are not available for the healthy group.

COMMENT 6:

Figure 1: Please give the reader the information, if this exemplary picture was obtained from a healthy or periodontitis patient. I would furthermore suggest to provide both, a healthy and a periodontitis example. Can you depict an endothelial cell as well in this exemplary picture?

RESPONSE:

This exemplary picture was taken from osteoclasts generated from adherent PBMCs of periodontitis patients (lines 196-197). This picture shows only a small part of the well, and therefore it is not representative of the whole number of osteoclasts per well. Therefore, it is unreasonable to show an additional picture for a healthy individual. Endothelial cells are not present in the picture because they were separated from PBMCs before osteoclasts generation.

COMMENT 7:

You should, either in the M&M section or at least in the Figure legend, give some information about the microscope and the used settings. 

RESPONSE:

This information was added to the legend of Figure 1 (lines 199-200).

COMMENT 8:

Fig. 4: Wouldn't a comparison with Co be more relevant compared to V-Co?

RESPONSE:

Since EMD is dissolved in acetic acid, we compared the data in EMD treated groups with V-Co. However, we recognize that such comparison is relevant only for the highest EMD concentration. Therefore, in the revised version, we included the data of statistical analysis in comparison to both Co and V-Co.

COMMENT 9:

Discussion: In the M&M sections, you write about patients with severe, generalized periodontal disease. Please discuss the quite low mean PPD of 3.6mm.

RESPONSE:

The quite low PPD value of 3.6 mm is explained by the fact that it represents the mean value of all measured sites (6 sites per tooth, all teeth). We emphasized this fact in the legend of Table 1 (lines 184-185). We would not like to discuss this point in the discussion to avoid the readers' distraction.

COMMENT 10:

Please discuss, how you made sure, that you see osteoclasts and no other cells, e.g. macrophages, in your wells. Have you performed some western blot or ELISA? 

RESPONSE:

This is a very important point. Macrophages can fuse into multinucleated cells. It is rather difficult to distinguish between osteoclasts and multinucleated giant cells only based on staining; some additional staining or test of bone-resorbing activity is required. This fact is discussed in the revised version as a potential limitation of our manuscript (see, lines 317-324).

COMMENT 11:

Line 294: As above: I would not seperate RANKL and OPG with a slash. I would suggest "RANKL- and OPG-pathway"

RESPONSE:

The sentence was corrected accordingly.

Reviewer 3 Report

The present cross-sectional study, concerning the influence of  the enamel matrix derivatives on osteoclasts formation from peripheral blood mononuclear cells in periodontitis vs periodontally healthy subjects is very interesting and innovative.

Methods are adequately described, the research design is appropriate and the inclusion/exclusion criteria are perfectly designed.

Manuscript structure is well organized. Both background and results are well presented. Discussion is interestingly written.

My only concerns about manuscript are about:

INTRODUCTION:

  • PMC or PBMC (line 42)
  • Please, specifies the references related to “both studies” (line 58).

MATERIALS AND METHODS:

  • Please, breifly describe the healthy volunteers group, as done for the patient group in lines 92-94
  • Please, move forward (to line 101) the sentence "Clinical history was recorded for all participants (personal data and medical history" (currently in lines 94-95).
  • With reference to the statement “The control group included periodontal healthy individuals presenting no radiographic bone loss, PPD ≤ 3 mm, and no  gingival inflammation” (lines 108-110), was also CAL used to discriminate periodontitis vs periodontally healthy subjects, especially considering the 2018 classification of periodontal and peri-implant diseases, previously applied for the staging (line 107)? If so, please, specify it in the text
  • Please, remove the sentence "Patients with systemic diseases and smokers were excluded (line 119) because redundant.

DISCUSSION:

  • Please, substitute “show” with “showed or have shown” (line 241)
  • Please re-phrase the periods in lines 273-279
  • With reference to the period “was observed already at 10 μg/ml of EMD, whereas in adherent 282 and non-adherent fraction, this effect was observed only at 100 μg/ml of EMD” (lines 282-283), please, add a comment to make the results more readable.

Author Response

Thank you very much for your positive and constructive feedback. Please, find below our response to the raised criticism points.

COMMENT 1:

PMC or PBMC (line 42)

RESPONSE:

This typo was corrected; it should be PBMC

COMMENT 2:

Please, specifies the references related to "both studies" (line 58).

RESPONSE:

The references were specified.

COMMENT 3:

Please, breifly describe the healthy volunteers group, as done for the patient group in lines 92-94

RESPONSE:

We have added the information about recruiting the control group (see, lines 113-115):

COMMENT 4:

Please, move forward (to line 101) the sentence "Clinical history was recorded for all participants (personal data and medical history" (currently in lines 94-95).

RESPONSE:

The sentence was moved.

COMMENT 5:

With reference to the statement "The control group included periodontal healthy individuals presenting no radiographic bone loss, PPD ≤ 3 mm, and no  gingival inflammation" (lines 108-110), was also CAL used to discriminate periodontitis vs periodontally healthy subjects, especially considering the 2018 classification of periodontal and peri-implant diseases, previously applied for the staging (line 107)? If so, please, specify it in the text

RESPONSE:

Periodontal health has corresponded to 2018 classification (Chapple et al.,2018); this fact is mentioned in the revised version (lines 114-115).

COMMENT 6:

Please, remove the sentence "Patients with systemic diseases and smokers were excluded (line 119) because redundant. 

RESPONSE:

The sentence was removed.

COMMENT 7:

Please, substitute "show" with "showed or have shown" (line 241)

ANSWER:

This sentence was improved.

COMMENT 8:

Please re-phrase the periods in lines 273-279

RESPONSE:

This paragraph was re-phrased (now lines 281-287).

COMMENT 9:

With reference to the period "was observed already at 10 μg/ml of EMD, whereas in adherent 282 and non-adherent fraction, this effect was observed only at 100 μg/ml of EMD" (lines 282-283), please, add a comment to make the results more readable.

RESPONSE:

This sentence was revised according to the Reviewer's suggestion.

Round 2

Reviewer 1 Report

The research work carried out by the authors is very interesting and innovative. The research is well developed.

Author Response

Thank you for your evaluation and positive feedback.

Reviewer 2 Report

Dear authors, dear editor,

the manuscript has been corrected and improved very thoroughly!

I would suggest to insert that periodontal health was confirmed by assessing a periodontal screening index and x-rays (OPG?) and to depict the maxima and minima of the PPD. PPDs are clearly not symmetrically distributed and hence it might be misleading for the reader (especially for dentists), when the only value for periodontitis patients is below 4 mm. 

Apart from that, the study is really well designed, written and discussed.

Author Response

We are thankful to this Reviewer for this remark. The additional information about the periodontal screening and x-ray analysis for the healthy group was added to the manuscript (see lines 117-118). Additionally, the range of PPDs for the healthy group and periodontitis patients was added to table 1. The changes regarding the second revision round are highlighted in blue.